# Intervention-Based Alignment of Code Search with Execution Feedback

**Hojae Han♠, Minsoo Kim♠, Seung-won Hwang♠,* Nan Duan▷, Shuai Lu▷**
♠Seoul National University, {stovecat, minsoo9574, seungwonh}@snu.ac.kr
▷Microsoft Research Asia, {shuailu, nanduan}@microsoft.com

## Abstract

One of the fundamental goals in code search is to retrieve a functionally correct code for a given natural language query. As annotating for correctness requires executing test cases (i.e. obtaining execution feedback), existing code search training datasets approximate text-code co-occurrences as positive execution feedback. However, this approximation may misalign models' retrieval decisions from ground-truth correctness. To address such limitation, we propose **C**ode **I**ntervention-based **R**einforcement **L**earning (CIRL) that perturbs training code to result in misalignment (i.e. code intervention), then tests models' decisions and corrects them with the execution feedback by reinforcement learning. The first technical contribution of CIRL is to induce the execution feedback from perturbation, without actual execution. Secondly, CIRL introduces structural perturbations using abstract syntax trees, going beyond simple lexical changes. Experimental results on various datasets demonstrate the effectiveness of CIRL compared to conventional approaches.

## 1 Introduction

Code search aims to retrieve code snippets that are most relevant to a given query. However, the notion of relevance often falls short of capturing the ultimate goal of *functional correctness*, focusing on whether the code would execute as the programmer intended (Ding et al., 2022). For instance, programmers would prefer $c$ (in Figure 1a) over $c'$ (in Figure 1b), as the entire "else" clause is missing in $c'$.

Ideally, ground truth labels for functional correctness should be obtained from *execution feedback*, collected after executing each code $c$ with a sufficient amount of test cases for each query $q$. We denote such execution feedback as $\mathrm{EF}(q, c)$. However, conventional training datasets, such as

---

*Corresponding author.

Query ($q$): *Find the longest increasing subsequence (LIS) of a given array.*

Code ($c$):
```python
from bisect import bisect_left

def find_LIS(array):
    dp = []
    for num in array:
        idx = bisect_left(dp, num)
        if idx == len(dp):
            dp.append(num)
        else:
            dp[idx] = num
    return len(dp)
```
**EF = 1  $R_\theta = 1$**

(a) A positive code $c$ to a query $q$ in conventional dataset.

Code ($c'$):
```python
from bisect import bisect_left

def find_LIS(array):
    dp = []
    for num in array:
        idx = bisect_left(dp, num)
        if idx == len(dp):
            dp.append(num)
    return len(dp)
```
**EF = 0  $R_\theta = 1$**

(b) A negative code $c'$ with misaligned $R_\theta$ with EF.

Figure 1: Example code snippets with the execution feedback (EF) from a running environment and the model's retrieval decisions ($R_\theta$) trained on a conventional code search dataset. The code line difference between the code pair is highlighted as blue.

CodeSearchNet (CSN; Husain et al., 2019), collect data from GitHub by regarding commented text descriptions as queries then labeling code snippets by query-code co-occurrences. This collection process can introduce an observation bias of lexical dissimilarity between positive and negative training snippets. Consequently, models can learn such observation bias, resulting in misaligned retrieval decisions with EF of returning a negative code with high lexical similarity as a false positive. For example, Figure 1b shows that a code search model

$\theta$, trained from a conventional dataset, decides to retrieve $c'$, or $R_\theta(q, c') = 1$, but this model decision is misaligned with the execution feedback, or $\text{EF}(q, c') = 0$.

The objective of this paper is to resolve the misalignment, i.e., $R_\theta \neq \text{EF}$. A straightforward solution is to expose models to misaligned code pairs and correct the model decisions by EF. In reinforcement learning (RL), this trial-and-error process is achieved by interacting with agents while perturbing input states, referred to as **intervention** (Lee and Bareinboim, 2020; Carta et al., 2023). We employ this intervention technique by perturbing the positive training code $c$ to generate its negative counterpart $c'$, where both code snippets yield the same model decision. Formally, $R_\theta(q, c') = R_\theta(q, c) = 1$ but $\text{EF}(q, c) > \text{EF}(q, c')$.

However, obtaining EF to correct decisions for each $c'$ is expensive, and to make matters worse, there are infinite possible perturbations satisfying the above misalignment condition. To identify a representative $c'$, existing solution known as counterfactual perturbation (Kaushik et al., 2020; Han et al., 2021; Choi et al., 2022; Chen et al., 2022), suggests to find $c'$ that is lexically closest to $c$, but with different EF. However, adopting this strategy would expose models solely to trivial syntax errors, such as omitting a colon from Figure 1a, causing a syntax error (and thus different EF). A realistic intervention, such as Figure 1a to Figure 1b, induces a larger lexical change, such as removing the "else" clause.

Our key technical contribution is a sample-efficient code intervention for RL. Our proposed approach, namely **C**ode **I**ntervention-based **R**einforcement **L**earning (CIRL), perturbs a positive code into negative with $\epsilon$ structural changes, which is designed to include the intuition of minimal lexical edit, but generalizes beyond to enfold structural similarity.

Specifically, we leverage the abstract syntax tree (AST) representation of the code, treating subtrees such as statements, clauses and expressions, as units of structural changes. A structural perturbation is done by masking out a subtree from a positive code then filling it out unless model prediction changes. We stress that CIRL subsumes counterfactual perturbation (when subtree is a leaf), and also adapts to the test distribution, by selecting subtrees with a realistic granularity (e.g. "else" clause in Figure 1a). Further, we ensure syntactic validity while replacing subtrees, to discourage exposures to trivial syntax errors.

We summarize our contributions as follows: First, we employ reinforcement learning with intervention that performs perturbation to simulate misaligned code without feedback from real code execution. Second, we propose CIRL, which utilizes AST information to inject $\epsilon$ structural changes to code to alter EF while preserving model decision. Third, our experimental results on various datasets show that CIRL effectively contributes to aligning code search with EF.

All our implementation and datasets are publicly available[1] for future research purposes.

## 2 Preliminary

In this section, we will begin by defining the code search task and addressing the misalignment issue in existing supervised code search training.

### 2.1 Code Search

For a universe of queries $\mathcal{Q}$ and code snippets $\mathcal{C}$, the ultimate goal of this task is to find functionally correct snippets for each query.[2] The functional correctness between a query $q \in \mathcal{Q}$ and a code $c \in \mathcal{C}$ can be confirmed by execution feedback (EF) from an environment using a set of test cases $\text{TC}_q$:

$$\text{EF}(q, c) = \begin{cases} 1, & \text{if } \forall(t_i, t_o) \in \text{TC}_q, c(t_i) = t_o, \\ 0, & \text{otherwise,} \end{cases}$$
(1)

where $(t_i, t_o) \in \text{TC}_q$ is a test input-output pair and $c(t_i)$ signifies running $c$ by feeding $t_i$. Our goal is to build a code search model $\theta$ making its retrieval decision $R_\theta \in \{0, 1\}$ to align with EF for any $q$ and $c$, i.e., $\forall q \in \mathcal{Q} \ \forall c \in \mathcal{C}, R_\theta(q, c) = \text{EF}(q, c)$.

### 2.2 Supervised Baseline

Ideally, a training dataset $D^*$ should comprise query-code pairs along with EF as labels, encompassing both positive and negative examples, i.e., $D^* = \{(q, c, \text{EF}(q, c)) | q \in \mathcal{Q}, c \in \mathcal{C}\}$. However, building such a dataset at scale is nontrivial due to the high costs and efforts involved in manually annotating test cases for each query, executing the code with test cases, and creating a secure

---

[1] https://github.com/stovecat/CIRL

[2] Although code search typically aims to retrieve relevant code snippets, pursuing functional correctness is valuable, e.g., in scenarios where generated codes are directly executed, as relevance is a subset of functional correctness.

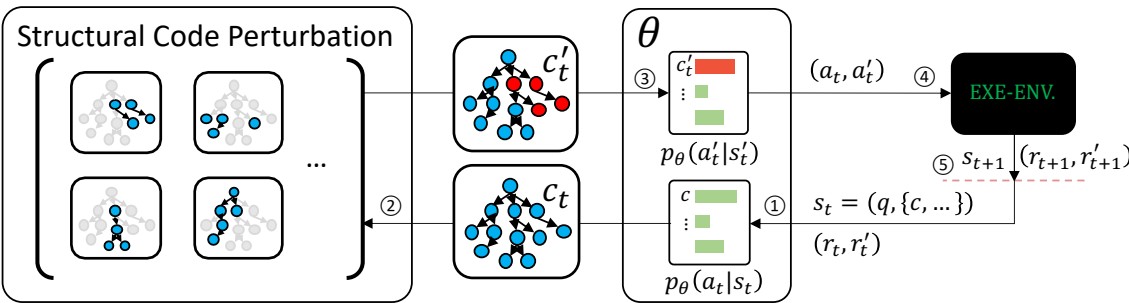

Figure 2: Overview of our reinforcement learning framework with code intervention. ① Given a state $s_t$ consists of a query and a set of candidate code snippets, a code search policy $\theta$ takes an action $a_t$ to retrieve a code $c_t$. ② CIRL perturbs the retrieved code $c_t$ into $c'_t$ with Abstract Syntax Tree (AST) information. ③ CIRL intervenes the state $s_t$ to $s'_t$ by replacing $c_t$ into $c'_t$ and makes $\theta$ take another action $a'_t$ from $s'_t$. If $c'_t$ is not retrieved, we go back to the second stage. ④ When $c'_t$ is retrieved by $a'_t$, though we expect to receive the EF from an execution environment as a reward—$r_{t+1} = \text{EF}(q, c_t)$ and $r'_{t+1} = \text{EF}(q, c'_t)$—CIRL can skip this costly stage by inducing the EF during perturbation. ⑤ $\theta$ is updated to align its decision with the EF.

environment for possibly unsafe code when executed (Chen et al., 2021; Li et al., 2022c). Due to this reason, a traditional training dataset such as CodeSearchNet (CSN; Husain et al., 2019) is gathered from GitHub by assuming that each function implementation $c$ and its corresponding commented description $q$ (i.e. $R(q, c) = 1$) can substitute EF: $D = \{(q, c, 1) | q \in \mathcal{Q}, c \in \mathcal{C}, R(q, c) = 1\}$.

In training time, negative code snippets for a query are sampled from other queries' positive snippets. However, these sampled negative snippets are both lexically and structurally dissimilar to positive ones. To illustrate, we examine that $\sim 97\%$ of positive-negative code pairs in CSN Python training set score very low in lexical/structural similarity.[3] As a consequence, dataset $D$ exhibits bias towards positive code, leading to functionally misaligned retrieval decisions by a model $\theta$ trained on $D$ when comparing positive code to lexically similar negative code. Figure 1 is an example of such bias—retrieving both $c$ and $c'$ despite their different EF due to a missing "else" clause.

## 3 Methodology

### 3.1 Contribution I: Reinforcement Learning with Intervention

We employ RL with code intervention to resolve the misalignment of model decisions and EF. Although RL behavior may resemble supervised or curriculum learning in specific scenarios where rules align with model decisions, our proposed RL

---

[3] $< 0.3$ CodeBLEU (Ren et al., 2020) scores, which considers both lexical and structural similarity.

abstraction enables to extend beyond these cases to situations where the RL-based model can choose to contradict (lesser-effective) curriculum rules.[4]

**RL Formulation.** In the process of retrieving positive code snippets from a pool of $M$ candidates in $D$ by a model $\theta$, we regard $\theta$ as a policy and formulate code search as a Markov decision process, which consists of following elements:

- **States**: Given a query $q$, the state $s_t$ in time step $t$ is defined as $(q, C_t)$ where $C_0$ is the set of $M$ candidate code snippets.

- **Actions**: $A(s_t)$ is a set of possible actions from the state $s_t$, and $a_t \in A(s_t)$ is a retrieval decision of a code $c_t \in C_t$, i.e., $R_\theta(q, c_t) = 1$.

- **Reward**: The reward of $a_t$ is defined as the EF of $c_t$, i.e., $\text{EF}(q, c_t)$.

- **Transition**: The transition from $s_t$ to $s_{t+1}$ by $a_t$ is done by removing $c_t$ from $C_t$.

- **Discount factor**: $\gamma \in [0, 1]$ is a discount factor to estimate the current value of future rewards, i.e., $G_t = \sum_{k=0}^{M-t-1} \gamma^k \text{EF}(q, c_{t+k})$.

**Code Intervention.** To resolve the misalignment, we intervene a retrieved code $c_t$ in step $t$ by replacing it into $c'_t$ such that $R_\theta(q, c'_t) = R_\theta(q, c_t) = 1$ but $\text{EF}(q, c_t) > \text{EF}(q, c'_t)$. The overall process of our RL with code intervention is illustrated in Figure 2. Formally, the pairwise policy gradient of $\theta$ in step $t$ from two actions $a_t$ and $a'_t$ respectively

---

[4] Refer to Appendix B for details.

retrieving code $c_t$ and $c'_t$ is,

$$\Delta\theta = (G_t - G'_t)(\nabla \log p_\theta(a_t|s_t) \\ -\nabla \log p_\theta(a'_t|s'_t)), \qquad (2)$$

where $p_\theta$ is the probability distribution of actions taken by $\theta$, and $s'_t$ is the state after intervention by replacing $c_t$ into $c'_t$. $G_t$ and $G'_t$ is the cumulative rewards starting from $t$ for $a_t$ and $a'_t$ respectively. As we replace the retrieved code $c_t$ into $c'_t$ to generate $s'_t$ from $s_t$, both states have the same next state, or, $s_{t+1} = s'_{t+1}$, thus $G_t - G'_t = \mathrm{EF}(q, c_t) - \mathrm{EF}(q, c'_t)$. Then Eq (2) becomes,

$$\Delta\theta = (\mathrm{EF}(q, c_t) - \mathrm{EF}(q, c'_t))(\nabla \log p_\theta(a_t|s_t) \\ -\nabla \log p_\theta(a'_t|s'_t)). \qquad (3)$$

This equation suggests that gradients can be calculated efficiently, in a single transition. Lastly, to avoid directly comparing the policy gradients from different states $s_t$ and $s'_t$, we approximate two states into a single virtual state $\bar{s}_t = (q, C_t \cup \{c'_t\})$. As $\theta$ can take both $a_t$ and $a'_t$ in $\bar{s}_t$, we only need to maximize $a_t$ in $p_\theta$:

$$\Delta\theta = (\mathrm{EF}(q, c_t) - \mathrm{EF}(q, c'_t))\nabla \log p_\theta(a_t|\bar{s}_t). \qquad (4)$$

From the perspective of Levine et al. (2020), our approach can be understood as an online off-policy RL, where $a'_t$ acts as an action produced by an off-policy and $\theta$ is the target policy. In this light, code intervention is akin to *importance sampling* that leverages predictions of the target policy to choose actions likely to be undertaken.

### 3.2 Contribution II: Structural Perturbation

However, computing $\Delta\theta$ in Eq (4) is still expensive, due to the following challenges: First, we need to compute $\mathrm{EF}(q, c')$ for each gradient calculation. Second, from the perspective of off-policy RL, we need to pursue sample efficiency, as the search space of $c'_t$ satisfying $R_\theta(q, c'_t) = R_\theta(q, c_t) = 1$ yet $\mathrm{EF}(q, c_t) > \mathrm{EF}(q, c'_t)$ is enormously broad.

A conservative approach is applying a minimal lexical edit, e.g., changing '+' operator to '-', to guarantee to alter the EF, keeping $R_\theta$ unchanged as two code snippets are lexically near-identical, which we denote as **lexical counterfactual perturbation** (Kaushik et al., 2020; Han et al., 2021; Choi et al., 2022; Chen et al., 2022). As we intended

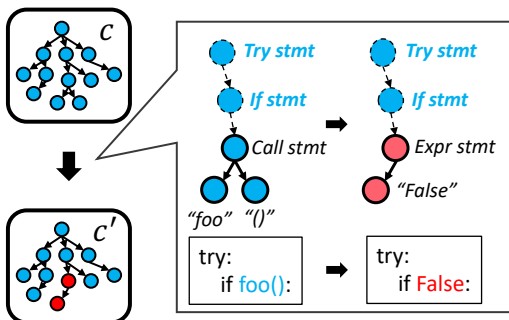

Figure 3: CIRL replaces subtrees that share the same ancestor AST node sequence ("try" statement → "if" statement) to avoid syntax errors.

the perturbation to flip EF from 1 to 0, we can label EF=0 for the perturbed code without incurring actual execution.

However, the counterfactual objective of minimizing lexical edits is too restrictive for compositional nature of code (Han et al., 2022). To illustrate, Figure 1a and 1b are lexically rather distant, but semantically close, or, parse tree structure is only $\epsilon$ away.

Inspired, we propose to relax lexical minimization to train more realistic EF-contrastive pairs, with the following goals:

- A pair should be free of syntax errors. Using lexical perturbation often incurs syntax errors such as omitting a semicolon.

- A pair must maintain contextual consistency. For example, when we change the if condition in Figure 3, lexical perturbation can break the context of 'try statement - if statement'.

- A pair should match the target distribution such as large lexical edits. Lexical perturbation with minimal edits results in deviate from the desired distribution.

Toward the goals, we propose subtree perturbation for augmenting a pair, ensuring the above criteria:

- **Controlled Syntax and EF Error**: We model code revisions utilizing Abstract Syntax Trees (ASTs), to consider subtrees like statements and clauses as elements for perturbations.[5]

---

[5] Empirical evidence suggests that less than 1% of the code snippets augmented by our method (sourced from the Codeforces training set) contain syntax errors and potential EF noises play negligible roles for our contrastive training objectives.

- **Maintaining Contextual Consistency**: As shown in Figure 3, the augmented code satisfies contextual consistency by matching the ancestor nodes during subtree replacement.

- **Target Distribution Matching**: The newly inserted subtrees, being derived from actual code snippets, match the target distribution.

Note that we can avoid expensive $EF(q, c')$ invocation, as it subsumes counterfactual perturbation as a special case where the subtree is a leaf node.

### 3.3 CIRL

We present CIRL by providing a summary of the steps involved in the code perturbation from the original code $c$ to $c'$. We begin with a positive query-code pair $(q, c)$ in the dataset $D$ where $EF(q, c) = 1$. We initialize a perturbation ratio $0 < \delta < 100\%$, which determines the number of nodes to be changed. This ratio is gradually increased through iterations using a schedule function $s$.[6] Note that a structural change of magnitude $\epsilon$ can lead to significant lexical modifications with a magnitude of $\delta$, impacting multiple leaf nodes.

During each iteration, we perform the following two steps and verify whether the model's decision on code $c$ is maintained for the perturbed code $c'$, i.e., $R_\theta(q, c) = R_\theta(q, c')$. If the decision is preserved, we calculate $\Delta\theta$ in Eq (4) using $c'$, increase $\delta$ using the schedule function $s$, and proceed with another iteration using the updated $\delta$. The iteration process terminates either when $R_\theta(q, c) \neq R_\theta(q, c')$ is satisfied or when the maximum number of iterations $n$ is reached.

**Step 1: Subtree Removal.** We convert $c$ into its AST representation. Then, based on the node perturbation ratio $\delta$, we randomly select a set of subtrees to remove. Each subtree's root node is a statement, clause, or expression. The total number of nodes for the selected subtrees is approximately equal to $\delta\%$ of the AST nodes for $c$.

**Step 2: Subtree Insertion.** For each removed subtree, we sample a new subtree from other code snippets in the dataset $D$. To maintain syntactic validity, we adopt a conservative guideline: we select a new subtree only if it shares the same ancestor AST node sequence with the removed subtree as depicted in Figure 3. To prevent name errors, we

uniformly sample names from the code $c$ for variables and functions in the injected subtrees. We ensure consistent replacements for repeated names. Finally, we convert the perturbed AST back into its code form, resulting in $c'$.

## 4 Experiments

### 4.1 Datasets

The evaluation is performed on three categories of benchmarks.

The first category involves large-scale public code competition datasets, namely Codeforces (Caballero et al., 2016) and CodeNet (Puri et al., 2021). These datasets consist of natural language descriptions paired with code submissions including Python. We use the cleansed version from CodeContests (Li et al., 2022c), filter examples not written in Python language, code/description longer than 512 tokens, and those which cannot be parsed into ASTs.[7] We train on Codeforces and conduct evaluations on both sets. To see the performance across diverse difficulty, we split CodeNet into 10 subsets by accept-ratios (the number of answer code snippets among submissions) as an additional evaluation set, to complement Codeforces that is relatively too easy (accept ratio is mostly 0.5 or higher).

The second category involves AdvTest (Lu et al., 2021), which evaluates the misalignment of positive code snippets that are not retrieved by code search models. This dataset is created by perturbing variable and function names while preserving EF from the test set of CodeSearchNet Python (CSN-Python; Husain et al., 2019) .

The third category focuses on CSN benchmark, which includes six different programming languages. The evaluation specifically examines the performance on the Python and Ruby subset of CSN, which is filtered from previous work (Guo et al., 2021).

Overall, these benchmark categories provide comprehensive evaluations of CIRL and its impact on aligning model decisions with EF across various datasets and scenarios.

### 4.2 Code Search Baselines

We consider two code search baselines in our experiments. First baseline is GraphCodeBERT (Guo et al., 2021), a popular code search model that uses data flow, a graph structure for representing the

---

[6]We empirically tune $\delta$ to adapt to the test distribution. Please refer to Appendix C.2 for more details.

[7]Detailed statistics are shown in Appendix D.

Metric: MRR

| Method | EF$(q, c')$ | | High-struct. sim. | Codeforces | | CodeNet | | | |
| --- | --- | --- | --- | --- | --- | --- | --- | --- | --- |
| | Zero-cost | Neg. | | Valid | Test | 0-10 | 10-20 | 20-30 | 30-40 |
| ContraCode | - | - | - | 84.24 | 83.33 | 44.21 | 46.83 | 57.51 | 54.13 |
| GraphCodeBERT | - | - | - | 90.58 | 73.71 | 39.24 | 48.02 | 53.48 | 56.14 |
| CI-ADV | ✓ | ✗ | ✓ | 75.65 | 74.54 | 34.54 | 42.79 | 48.18 | 49.58 |
| CI-PLM | ✗ | ✓ | △ | 88.91 | 75.52 | 45.65 | 47.82 | 60.14 | 67.17 |
| CI-HUMAN | ✗ | ✓ | ✓ | 93.48 | 76.40 | 44.68 | **60.17** | 62.31 | 68.84 |
| CIRL | ✓ | ✓ | ✓ | 92.39 | 79.90 | 54.99 | 55.98 | 63.23 | 66.09 |
| + CI-HUMAN | ✗ | ✓ | ✓ | **96.74** | 79.02 | **56.99** | 57.86 | **68.36** | **71.04** |
| - Code Intervention | ✗ | ✓ | ✓ | 86.96 | **85.75** | 43.86 | 51.89 | 55.53 | 64.63 |

| Method | EF$(q, c')$ | | High-struct. sim. | CodeNet | | | | | |
| --- | --- | --- | --- | --- | --- | --- | --- | --- | --- |
| | Zero-cost | Neg. | | 40-50 | 50-60 | 60-70 | 70-80 | 80-90 | 90-100 |
| ContraCode | - | - | - | 59.39 | 70.16 | 73.78 | 71.44 | 73.45 | 93.93 |
| GraphCodeBERT | - | - | - | 65.29 | 68.16 | 73.24 | 76.34 | 74.04 | 94.65 |
| CI-ADV | ✓ | ✗ | ✓ | 56.27 | 66.15 | 68.25 | 73.19 | 70.98 | 93.67 |
| CI-PLM | ✗ | ✓ | △ | 66.61 | 74.35 | 81.31 | 82.77 | 81.24 | 98.11 |
| CI-HUMAN | ✗ | ✓ | ✓ | 77.34 | 77.77 | 83.26 | 86.59 | 85.54 | 96.86 |
| CIRL | ✓ | ✓ | ✓ | 78.01 | 80.64 | 87.50 | 90.81 | 87.99 | 98.11 |
| + CI-HUMAN | ✗ | ✓ | ✓ | **79.32** | **82.44** | **87.75** | **90.94** | 89.79 | **99.06** |
| - Code Intervention | ✗ | ✓ | ✓ | 72.92 | 77.97 | 83.61 | 84.94 | **89.97** | 93.87 |

Table 1: Results on Codeforces and CodeNet. Zero-cost and Neg. columns below EF$(q, c')$ respectively stands for whether the method requires zero-cost to obtain EF of intervened data, and whether intervened codes receive negative EF. High-struct. sim. column signifies whether intervened codes share high structural similarity with original codes, where a triangle for CI-PLM means that the high similarity is not ensured as language models are hard to be fully controlled in their generation process. '- Code Intervention' is an ablation study that employs a simple curriculum contrastive learning instead of reinforcement learning with code intervention.

relation among variables–declaration, assignment, usage, and removal. Second baseline is Contra-Code (Jain et al., 2021), which conducts contrastive pretraining on conventional datasets. To minimize truncation, we use a bi-encoder for encoding the description and code separately, following Guo et al. (2021) and Jain et al. (2021).[8] We obtain the last layer's representation of the [CLS] token for each query and code, then compute the prediction score.[9]

## 4.3 Code Intervention Baselines

We compare CIRL with different code augmentation approaches as code intervention baselines instead of our structural perturbation.[10] Note that

we compare CI-HUMAN and CI-PLM on Code-forces and CodeNet, because of the prerequisites in both baselines– the existence of test cases and human generated negative code snippets.

**CI-HUMAN** This baseline utilizes human efforts by generating additional code snippets that receive negative EF. Unlike conventional distribution such as CSN, Codeforces provides not only the set of positive code for each query, but also the set of negative code for each query. This approach additionally utilizes the latter set for code intervention.

**CI-PLM** Following Inala et al. (2022), we can augment negative code snippets generated by code generation models instead of expensive human efforts. We use GPT-Neo 2.7B (Black et al., 2021) finetuned on APPS (Hendrycks et al., 2021) dataset to generate snippets in Codeforces training set, then calculate the EF for each generated code.

---

[8]Note that CIRL maintains a model-agnostic nature, as it acknowledges the potential enhancement of cross-encoder models by intervened data, such as iteratively sampled hard negatives in dense text retrieval, as demonstrated in Zhang et al. (2022).

[9]Refer to Appendix C.1 for implementation details.

[10]Refer to Appendix C.2 for implementation details.

Metric: MRR

| Model | Maximum Train Epochs | AdvTest |
|---|---|---|
| ContraCode | 10 | **22.2** |
| CIRL | 10 | 22.1 |
| GraphCodeBERT | 2 | 35.2[†] |
| GraphCodeBERT | 10 | 37.1 |
| CI-ADV | 10 | 32.3 |
| CIRL | 10 | **39.0** |

Table 2: Results on AdvTest: All results were optimized with early stopping, tailored to performance on the validation set. We have omitted CI-PLM and CI-HUMAN since the training set (CSN) lacks test cases and human annotations. The result for [†] is reported in Wang et al. (2021).

**CI-ADV**    Like in AdvTest, code intervention may tackle misaligned model decisions caused by positive code that are not retrieved, by semantic-preserving perturbation of each positive code $c$ from $D$ to $c'$ such that $R_\theta(q, c) > R_\theta(q, c')$ but $EF(q, c') = EF(q, c) = 1$. Following Bui et al. (2021) and Lu et al. (2022), heuristic rules such as the replacement of variable and function names, or the insertion of dead code lines can be seen as defining such adversarial perturbation.

### 4.4   Main Results

For evaluation, we used Mean Reciprocal Rank (MRR), a standard code search metric from previous studies (Feng et al., 2020; Guo et al., 2021; Lu et al., 2021; Wang et al., 2021), with respect to the ground-truth of ranking the correct answer as the top.

**Codeforces and CodeNet**    Table 1 presents improved MRR scores for evaluating the alignment of model decisions with EF using different code intervention approaches. Among the low-cost methods, CIRL outperformed ContraCode, GraphCodeBERT, and CI-ADV. Even when compared to the costly methods, CIRL still significantly outperformed CI-PLM and generally outperformed CI-HUMAN. Regarding the code difficulty, CI-Human enhanced MRR by over 2% in difficult CodeNet ranges (0-10 to 30-40), but less in easier ranges (40-50 to 90-100). Conversely, CIRL consistently boosted MRR across all code difficulties, showcasing its widespread efficacy. Additionally, CIRL combined with CI-HUMAN, achieved the best performance, showing the augmentation from

Metric: MRR

| Model | Maximum Train Epochs | CSN-Python |
|---|---|---|
| ContraCode | 10 | 59.9 |
| CIRL | 10 | **62.2** |
| GraphCodeBERT | 10 | **69.2[‡]** |
| CI-ADV | 10 | 64.1 |
| CIRL | 10 | 68.3 |

Table 3: Results on CSN-Python: All results were optimized with early stopping, tailored to performance on the validation set. We have omitted CI-PLM and CI-HUMAN since CSN-Python lacks test cases and human annotations. The result for [‡] is reported in Guo et al. (2021).

Metric: MRR

| Model | Maximum Train Epochs | CSN-Ruby |
|---|---|---|
| GraphCodeBERT | 10 | 70.3[‡] |
| CIRL | 10 | **72.0** |

Table 4: Results on CSN-Ruby: All results were optimized with early stopping, tailored to performance on the validation set. The result for [‡] is reported in Guo et al. (2021).

CIRL can complement human efforts. Lastly, the '- Code Intervention' ablation replaced RL with a basic curriculum contrastive learning, following settings in Appendix B. Its underperformance compared to CIRL highlights our RL mechanism's effectiveness.

**AdvTest**    To evaluate whether CIRL contributes to aligned model decisions for adversarially perturbed positive code, we applied CIRL on CSN-Python training set and evaluate on AdvTest. The training epochs were increased from 2 to 10, resulting in improved GraphCodeBERT performance. The results in Table 2 confirm that applying CIRL on ContraCode maintained performance, while it enhanced performance on GraphCodeBERT, indicating that CIRL is reliable and potentially beneficial for correcting misaligned decisions in adversarial positive code. In contrast, CI-ADV exhibited a drop of approximately 5% compared to GraphCodeBERT, which is further discussed in Section 5.4.

**CSN**    The results on the conventional distribution, where observation bias is strongly correlated with the label, are presented in Table 3. CIRL enhanced

| Metric: MRR | CodeNet + Structural Perturbation by CIRL | | | | |
|---|---|---|---|---|---|
| Model | 0-10 | 10-20 | 20-30 | 30-40 | 40-50 |
| ContraCode | 37.32 -6.89 | 39.30 -7.53 | 44.99 -12.52 | 35.34 -18.79 | 38.40 -20.99 |
| GraphCodeBERT | 32.79 -6.45 | 38.99 -9.03 | 40.25 -13.23 | 41.60 -14.54 | 43.65 -21.64 |
| +CI-HUMAN | 42.34 -2.34 | 57.30 -2.87 | 55.04 -7.27 | 60.82 -8.02 | 70.75 -6.59 |
| CIRL | 54.99 -0.00 | 55.98 -0.00 | 63.23 -0.00 | 66.09 -0.00 | 78.01 -0.00 |
| + CI-HUMAN | 56.99 -0.00 | 57.86 -0.00 | 68.36 -0.00 | 71.04 -0.00 | 79.32 -0.00 |

Table 5: Test time code augmentation by our structural perturbation to CodeNet. The red colored numbers are MRR drops after augmentation of intervened codes by CIRL. Note that CIRL is trained without the test set.

the MRR score on ContraCode and showed only a marginal decrease on GraphCodeBERT. Additionally, Table 4 demonstrates CIRL's effectiveness across programming languages, showcasing results on Ruby code snippets.

## 5 Discussion

### 5.1 Is CIRL Perturbation Sample-efficient?

We conducted a stress test by applying CIRL on 5 test subsets of CodeNet (0.0-0.1 to 0.4-0.5).[11] The results in Table 5 demonstrate that both Contra-Code and GraphCodeBERT experienced drops in MRR scores after augmenting the intervened code snippets, indicating confusion between intervened negative code and positive code. Although CI-HUMAN reduced this confusion, the model still suffered from MRR drops. However, applying CIRL successfully resolved this confusion and prevented MRR drops.

### 5.2 Does CIRL Generalize Over Counterfactual Perturbation?

We examined the impact of CIRL allowing for more structural changes, compared to counterfactual perturbation with minimal lexical changes. To ensure a fair comparison, we implemented a COUNTERFACTUAL baseline from CIRL by fixing the perturbation ratio $\delta$ to a small value ($\delta = 2$) to focus on small lexical changes. Figure 4 presents a comparison of GraphCodeBERT, COUNTERFAC-TUAL, and CIRL on Codeforces and CodeNet benchmarks. As expected, COUNTERFACTUAL exhibited lower MRR scores compared to CIRL, and even lower than GraphCodeBERT in some test sets.

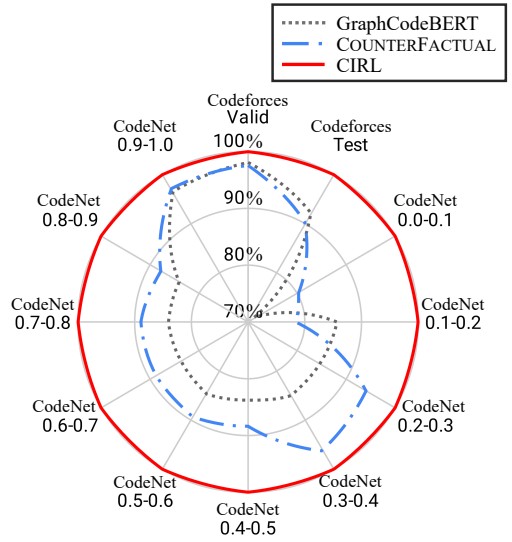

Figure 4: Relative MRR performance to CIRL. COUN-TERFACTUAL approach which intervenes code with a minimal lexical edit is suboptimal.

### 5.3 Does CIRL Generalize Over Different Programming Languages?

Table 4 shows that CIRL successfully improved the MRR performance on CSN-Ruby. Though our AST perturbation mechanism may look more complex than lexical perturbation, it straightforwardly generalizes to other languages, as AST parser (e.g., *tree-sitter*) easily applies to other languages for extracting a language-generic representation to perturb.

### 5.4 Can Augmentation Negatively Affect Performance?

It may seem counter-intuitive that one of the existing augmentation approaches, CI-ADV, negatively affects the MRR scores throughout our experiments. We speculate that CI-ADV relies on heuristic rules for perturbation, which inherently have limitations

---
[11]Refer to Appendix C.3 for details.

in terms of the coverage of perturbations in order to preserve EF. For instance, augmenting code perturbation by variable renaming results in high lexical and structural similarity with the original code. Consequently, this augmentation may amplify the observation bias from the original code, leading to more misaligned decisions in trained models.

## 5.5 Can LLMs Align better with EF?

We explore whether general-purpose large language models, benefiting from enormous parameter scales, can detect misalignments. To confirm this, we use GPT-3.5 (OpenAI, 2022) to evaluate query-code pairs from CodeNet 0.0-0.1 and determine whether the code is correct for the given query, in the following two settings: (1) zero-shot; (2) in-context learning (ICL) by providing a positive $(q, c)$ and a negative pair $(q, c')$ randomly sampled from the Codeforces training set.[12]

**Fail to Align Human Perturbations.** GPT-3.5 inaccurately identified negative code snippets by CI-HUMAN as false positives, performing worse than random guessing of 50% (see the 'Negative Codes (CI-HUMAN)' graphs in Figure 5).

**Fail to Align Subtree Removals.** To illustrate the weakness of GPT-3.5 for negative code with subtree removals, we implemented $CIRL_{RM}$ doing the removal step only. GPT-3.5 failed to identify omitted implementations such as handling edge cases[13], as shown in Figure 5 that 'Negative Codes ($CIRL_{RM}$)' graphs were lower than random guessing (50%).

**Succeed When Code Lines are Replaced.** GPT-3.5 showed $100\%$ aligned decisions with EF for negative code snippets consisting of replaced subtrees, shown as 'Negative Codes (CIRL)' graphs in Figure 5. GPT-3.5 often explained the reason for its decision as the presence of unnecessary or unrelated code lines.[13]

**In-Context Learning Fails.** A naive solution of employing ICL with positive and negative examples did not improve the alignment, as contrasted by zero-shot and ICL groups in Figure 5. Against our expectation, it even increased misalignment for negative code by CI-HUMAN and $CIRL_{RM}$. We leave the alignment of LLMs' decisions with EF for future work.

[12]Refer to Appendix C.4 for details.
[13]Example outputs are shown in Appendix F.

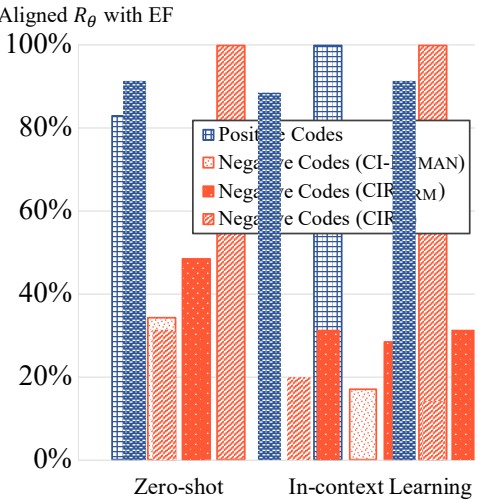

Figure 5: Alignment accuracy (i.e. 100% if $R_\theta$ = EF for all $R_\theta$) of GPT-3.5 on CodeNet 0.0-0.1. For in-context learning, we additionally gave a query and its positive-negative code snippets sampled from Codeforces training set.

## 6 Conclusion

This paper has explored the issue of misalignment of model decisions and EF due to observation bias in traditional training datasets. To overcome this limitation, we have introduced a novel RL framework with code intervention called CIRL. The primary contribution of CIRL is to expose models to misaligned code snippets, which can be subsequently corrected through EF. CIRL is sample efficient by utilizing ASTs to simulate structural perturbations for code intervention, allowing us to bypass actual execution for EF. Extensive experimental results on various datasets demonstrate that CIRL enhances the alignment of model decisions and EF in code search compared to conventional approaches.

## Acknowledgement

This work was partially supported by Microsoft Research Asia, and Electronics and Telecommunications Research Institute (ETRI) grant funded by ICT R&D program of MSIT/IITP (2022-0-00995, Automated reliable source code generation from natural language descriptions).

## Limitations

Despite our achievements, there are following limitations in this work. First, we mainly tackled Python language, appending with Ruby. Second, while CIRL is model-agnostic, we applied it to

ContraCode and GraphCodeBERT, conducting preliminary analysis on LLMs, and leaving extensive application on LLMs for future work.

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

# A Related Work

## A.1 Deep Code Search Models

Recently, deep learning models have shown dramatic improvements in the conventional code search task. Early approaches (Gu et al., 2018; Feng et al., 2020) directly mimic NLP architectures and objectives. Later approaches additionally utilize code specific information such as data flow (Guo et al., 2021), abstract syntax tree (AST; Wang et al., 2021; Guo et al., 2022, and template (Han et al., 2022). For pretraining, several approaches (Jain et al., 2021; Wang et al., 2021; Guo et al., 2022; Li et al., 2022b) employ contrastive learning of semantically related code-code and text-code pairs. Instead of random sampling, existing works (Li et al., 2022a; Shi et al., 2023) have proposed negative code sampling methods in expense of extensive model predictions or additional memory usage.

**Our distinction.** Existing approaches are trained by (self-)supervised learning in conventional datasets where execution feedback (EF) is approximated by text-code coocurrences, suffering from misalignment of model decisions with EF. Our distinction is to employ reinforcement learning and intervene code to test and correct misaligned model decisions. Regarding negative code sampling, our approach plays a similar role as hard negative mining, all while circumventing the need for extensive computational resources or memory usage.

## A.2 Reinforcement Learning in IR and Code Generation

**Reinforcement Learning to Rank.** In information retrieval (IR) and recommender system, which are related to code search, recent approaches have been applied reinforcement learning (RL), formulating document retrieval and item recommendation as markov decision process (MDP). In early approaches like PPG (Wei et al., 2017) a policy estimates the absolute rank score of each candidate (i.e. pointwise). Inspired by pairwise learning to rank (Liu, 2009), later approaches (Zhao et al., 2018; Xu et al., 2020; Hu et al., 2018) use pairwise loss between candidates sharing certain features with different feedback, improving sample efficiency.

**Reinforcement Learning on Code Generation.** Several approaches have employed RL to generate code snippets. CodeRL (Le et al., 2022)

uses an actor-critic algorithm that gives rewards for policy gradients by a trained reward model. PPOCoder (Shojaee et al., 2023) utilizes Proximal Policy Optimization (PPO) to reduce syntax errors and improve functional correctness.

**Our distinction.** Unlike IR, recommender system, and code generation domains, traditional code search training sets do not contain misaligned code pairs. Our distinction is to generate misalignment pairs with intervention, by perturbing a given positive code to a negative code while not changing model decision.

## A.3 Synthetic Code Augmentation

To automatically supplement training code distribution, existing approaches (Bui et al., 2021; Yang et al., 2022) utilize semantic-preserving perturbations by changing semantically ineffective lexicons (e.g., variable and function names). Other approaches target software vulnerability by semantic-preserving perturbations to conduct adversarial attacks (Jha and Reddy, 2023), or learning to capture the semantics (Ding et al., 2022). Meanwhile, several approaches like CODERANKER (Inala et al., 2022) augment generated code snippets to train code search models.

**Our distinction.** Unlike existing methods, we directly target to code intervention in RL to align code search with EF, while managing both sample efficiency and zero-cost for EF calculation.

## A.4 Counterfactual Text Augmentation

In NLP, early approaches (Kaushik et al., 2020) pair each instance with its label-flipped augmentation, obtained from perturbing labels by human efforts with $\epsilon$ lexical changes. Later approaches propose automatic syntheses to alleviate human efforts (Han et al., 2021; Paranjape et al., 2022; Choi et al., 2022; Calderon et al., 2022).

**Our distinction.** Applying $\epsilon$ lexical changes to flip EF for code intervention is not sufficient as it only exposes to trivial lexical errors such as forgetting colon from Figure 1a, thus fails to generalize in misaligned decisions over larger lexical changes like Figure 1b. Instead, we produce $\epsilon$ structural changes, which subsume $\epsilon$ lexical changes as a special case of replacing a single leaf node, thus broaden model exposures for generalization.

| Metric: MRR | |
|---|---|
| Method | Rank Order |
| Contrastive Learning w/ Curriculum | $\text{NEG}_{diff}$ (Top 1) > $\text{NEG}_{sim}$ (Top 6) > POS (Top 7) |
| CIRL | POS (Top 1) > $\text{NEG}_{diff}$ (Top 7) > $\text{NEG}_{sim}$ (Top 15) |

Table 6: The rank order of three code snippets for each method. Each code is respectively shown in Table 7, 8, and 9.

## B  Motivation of RL framework

Both supervised learning and curriculum learning can be viewed as specific instances of the RL framework. Thus, it shouldn't be surprising that in certain fortuitous scenarios when $R_\theta(q, c) = R_\theta(q, c')$ for every $c'$, the RL framework may resemble to them. However, this does not undermine the value of our RL framework. To illustrate, curriculum learning utilizes a predetermined rule for data selection during augmentation (like measuring AST perturbation levels). Our proposed RL framework encompasses such fortuitous scenarios where the rule strongly correlates with agent decision, making both RL and curriculum learning optimal.

However, our framework extends beyond these specific cases, into general scenarios when (a) agent disagrees with the rules from curriculum learning, contributing (b) disagreements to gains:

**(a) Agent-Curriculum Disagreements.** We sampled 100 positive code snippets from the Codeforces (Caballero et al., 2016) training set and corresponding generated negative snippets with five different perturbation ratios (same as in Appendix C.2). A Spearman rank correlation analysis between selection orderings of these negative snippets by curriculum and by RL with intervention revealed that 17% of the rank correlations are below 0.5. This underscores that curriculum learning and RL-based intervention often have differing data selection preferences.

**(b) Disagreements Contribute to Gains.** An analysis of cases where curriculum learning faltered (i.e., top 1 ranked code is negative) on the CodeNet (Puri et al., 2021) 0.0-0.1 test set revealed frequent rank discrepancies. For instance, the top 1 ranked negative code typically shared only one line with the positive code, whereas the negative code with the most line overlaps had 19 lines in common with its positive counterpart. Table 6 provides the qualitative evidence. Here, CIRL differs from the curriculum baseline by intervening code based on

target agent predictions. From the online off-policy RL perspective mentioned in Section 3.1, this data selection driven by the target agent can enhance the quality of decisions made by the agent during testing.

## C  Implementation Detail

### C.1  Code Search Baselines

**GraphCodeBERT**  For all scenarios except CSN-Ruby, we finetuned the pretrained checkpoint[14] of GraphCodeBERT (Guo et al., 2021) on the CSN (Guo et al., 2021) Python training set for 10 epochs using AdamW optimizer, a 2e-5 learning rate, max token lengths 128 (description) and 256 (code), a max data flow length of 64, and a batch size of 32. The total training time was 1.5 days with 4 NVIDIA GeForce RTX 3090 (24GB) GPUs. In Codeforces (Caballero et al., 2016) training set, we further fine-tuned GraphCodeBERT using 16 NVIDIA Tesla V100 (32GB) GPUs for 3 days with the following settings: using AdamW optimizer, a 2e-5 learning rate, a batch size of 480, a max training epoch of 10, max token lengths 512 for both description and code, and a max data flow length of 64.

**ContraCode**  We finetune the pretrained checkpoint of ContraCode (Jain et al., 2021) on Codeforces training set using 8 NVIDIA GeForce RTX 3090 (24GB) GPUs for 3 days with the same setting of GraphCodeBERT, except for the batch size of 96 and not using data flow. For CSN-Python and CSN-Ruby, we used the same setting of Graph-CodeBERT but not using data flow.

### C.2  Code Intervention Approaches

Throughout all approaches, we use GraphCode-BERT as a code search policy.

**CIRL**  In Codeforces training set, CIRL intervenes each positive code with maximum $n = 5$ iterations, where the initial perturbation ratio $\delta$

---

[14]https://huggingface.co/microsoft/graphcodebert-base

is initialized by $\delta = 2$ and updated by a schedule function $s(\delta, \mu) = \delta^\mu$ where $\mu \in [1, 2, ..., n]$ is the current iteration. The structural element is one of the following AST subtrees: future import statement, import statement, import from statement, print statement, assert statement, expression statement, return statement, delete statement, raise statement, pass statement, break statement, continue statement, global statement, nonlocal statement, exec statement, if statement, for statement, while statement, try statement, with statement, function definition, class definition, decorated definition, elif clause, else clause, except clause, with clause, and block. In CSN training set, CIRL conducts a single intervention per code ($n = 1$), where the perturbation ratio is set by $\delta = 32$. For other hyperparameters, we use the same values with Graph-CodeBERT, except for changing the batch size to 40 training with 8 NVIDIA GeForce RTX 3090 (24GB) GPUs for 10 days on Codeforces training set.

**CI-HUMAN**   We augment 1 negative code for each positive code to avoid overfitting, as the size of negative code snippets is 30.70% of that of positive snippets in Codeforces training set.

**CI-PLM**   We use GPT-Neo 2.7B (Black et al., 2021) finetuned on APPS (Hendrycks et al., 2021) dataset to generate 5 code snippets per problem description in Codeforces training set, then calculate the execution feedback (EF) for each synthetic code using the provided test cases from CodeContests (Li et al., 2022c). We train this baseline on the same setting with our approach, including the batch size of 40.

**CI-ADV**   To implement CI-ADV, we use the available implementation from Lu et al. (2022) for variable/function renaming and and dead code insertion. In both Codeforces and CSN training sets, we augment the same number of iterations for each positive code, along with the same training configuration.

### C.3   Test Time Code Augmentation by CIRL

We intervene with a single iteration for each positive code in 5 CodeNet (Puri et al., 2021) subsets (0.0-0.1 to 0.4-0.5), where each of the perturbation ratio $\delta = 5, 15, 25, 35,$ and $45$ is set by the average of min-max accept ratio for each subset.

### C.4   Misalignment in LLMs

We use GPT-3.5-turbo-0613 (OpenAI, 2022)[15], with the top_p as 1 and the temperature as 0 for reproducibility. Appendix F shows examples of input prompts for both zero-shot and in-context learning settings.

**CIRL$_{RM}$**   For each code, we remove a single structural element, where element is one of the following AST subtrees that do not incur syntax errors when removed: print statement, assert statement, if statement, for statement, while statement, try statement, with statement, function definition, class definition, else clause, except clause, and with clause.

---

[15]https://platform.openai.com/docs/model-index-for-researchers

```
from sys import stdin
from operator import attrgetter
readline = stdin.readline

def norm(a):
    return a.real * a.real + a.imag * a.imag

def closest_pair(p):
    if len(p) <= 1:
        return float('inf')
    m = len(p) // 2
    d = min(closest_pair(p[:m]), closest_pair(p[m:]))
    p = [pi for pi in p if p[m].imag - d < pi.imag < p[m].imag + d]
    return brute_force(p, d)

def brute_force(p, d=float('inf')):
    p.sort(key=attrgetter('real'))
    for i in range(1, len(p)):
        for j in reversed(range(i)):
            tmp = p[i] - p[j]
            if d < tmp.real:
                break
            tmp = abs(tmp)
            if d > tmp:
                d = tmp
    return d

def main():
    n = int(readline())
    p = [map(float, readline().split()) for _ in range(n)]
    p = [x + y * 1j for x, y in p]

    p.sort(key=attrgetter('imag'))
    print(':.6f'.format(closest_pair(p)))
main()
```

Table 7: **POS**: A correct solution (EF=1) for a test query in CodeNet 0.0-0.1.

```
from sys import stdin
import operator
readline = stdin.readline

def norm(self):
    return abs(self)

def closest_pair(p):
    m = 0
    p.sort(key=operator.attrgetter('real'))
    d = float('inf')
    for i in range(1, len(p)):
        for j in reversed(range(m, i)):
            tmp = p[i] - p[j]
            if d < tmp.real:
                m = j + 1
                break
            tmp = abs(tmp)
            if d > tmp:
                d = tmp
    return d

from itertools import combinations

def brute_force(p):
    return min(abs(p[i] - p[j]) for i, j in combinations(range(len(p)), 2))

def main():
    n = int(readline())
    p = [map(float, readline().split()) for _ in range(n)]
    p = [x + y * 1j for x, y in p]

    # print('{:.6f}'.format(brute_force(p)))
    print('{:.6f}'.format(closest_pair(p)))
main()
```

Table 8: **NEG**$_{sim}$: An incorrect solution (EF=0) for a test query in CodeNet 0.0-0.1. 55.2% of code line overlaps with POS in Table 7.

```
import numpy as np
n = int(input())
p = []
for i in range(n):
    p.append(tuple(map(float,input().split())))
d = []
min = 999999999999
for i in range(n-1):
    for j in range(i+1,n):
        a = ((p[j][0]-p[i][0])**2+(p[j][1]-p[i][1])**2)
        if a < min:
            min = a

print(round(np.sqrt(min),11))
```

Table 9: **NEG**$_{diff}$: An incorrect solution (EF=0) for a test query in CodeNet 0.0-0.1. 0% of code line overlaps with POS in Table 7.

# D  Dataset

| Dataset Type | Dataset Name | No. Descriptions | No. Answer Codes | No. Wrong Answer Codes | Accept Ratio (Answers/Total Candidates) |
|---|---|---|---|---|---|
| Training dataset | Codeforces (Train) | 2,734 | 842,296 | 258,551 | 76.51% |
| | Codeforces (Valid) | 23 | 2,078 | 1,578 | 56.84% |
| | Codeforces (Test) | 34 | 2,336 | 1,914 | 54.96% |
| | CodeNet 0.0-0.1 | 35 | 578 | 9,514 | 5.73% |
| | CodeNet 0.1-0.2 | 58 | 17,127 | 87,466 | 16.37% |
| | CodeNet 0.2-0.3 | 87 | 47,025 | 138,043 | 25.41% |
| | CodeNet 0.3-0.4 | 115 | 88,936 | 167,185 | 34.72% |
| Evaluation dataset | CodeNet 0.4-0.5 | 151 | 132,689 | 161,173 | 45.15% |
| | CodeNet 0.5-0.6 | 137 | 161,632 | 134,858 | 54.52% |
| | CodeNet 0.6-0.7 | 159 | 235,737 | 126,197 | 65.13% |
| | CodeNet 0.7-0.8 | 232 | 427,723 | 141,323 | 75.16% |
| | CodeNet 0.8-0.9 | 169 | 317,505 | 61,476 | 83.78% |
| | CodeNet 0.9-1.0 | 53 | 16,914 | 1,359 | 92.56% |

Table 10: Statistics for Codeforces and CodeNet.

# E  CIRL: Case Study

Figure 6: Examples of original (left) and intervened (right) code snippets by CIRL.

## F GPT-3.5: Case Study

Please answer whether the following code is the correct implementation of the problem:

**Problem**: "Score : 700 points Problem Statement N cells are arranged in a row. Some of them may contain tokens. You are given a string s that consists of 0 s and 1 s. If the i-th character of s is 1, the i-th cell (from left) contains a token. Otherwise, it doesn't contain a token. Snuke wants to perform the following operation as many times as possible. In each operation, he chooses three consecutive cells. Let's call the cells X, Y, Z from left to right. In order for the operation to be valid, both X and Z must contain tokens and Y must not contain a token. Then, he removes these two tokens and puts a new token on Y. How many operations can he perform if he performs operations in the optimal way?

Constraints

$1 \leqslant N \leqslant 500,000$

$|s| = N$

Each character in s is either 0 or 1.

Input

Input is given from Standard Input in the following format:

N

s

Output

Print the answer.

Sample Input 1

7

1010101

Sample Output 1

2

For example, he can perform two operations in the following way: Perform an operation on the last three cells. Now the string that Perform an operation on the first three cells. Now the string that represents tokens becomes 0100010. Note that the choice of operations matters. For example, if he chooses three cells in the middle first, he can perform no more operations.

Sample Input 2

50

10101000010011011110001001111110000101010111100110

Sample Output 2

10"

**Code**: "import re
def update(d, k, v):
    if k in d:
        d[k] = max(d[k], v)
    else:
        d[k] = v
rep_zeros = re.compile('0{2,}')
n = input()
s = input().strip('0')
ans = 0
for ps in re.split(rep_zeros, s):
    fs = ps.split('0')
    lfs = len(fs)
    if lfs == 1:
        continue
    ls = list(map(len, fs))
    pl = {ls[0]: 0}
    for cc in ls[1:]:
        nl = {}
        for pc, a in pl.items():
            update(nl, cc, a)
            if pc == 0:
                continue
            if cc == 1:
                update(nl, 0, a + pc)
            else:
                update(nl, 0, a + cc)
                update(nl, 1, a + cc - 1)
                if cc > 1:
                    update(nl, cc - 1, a + pc)
        pl = nl
    ans += max(pl.values())
print(ans)"

Yes, the given code is the correct implementation of the problem. It correctly solves the problem by finding the maximum number of operations that can be performed in the optimal way. The code uses regular expressions to split the string into consecutive sequences of 1s and then calculates the maximum number of operations that can be performed on each sequence. Finally, it sums up the maximum number of operations for all sequences and prints the result.

Table 11: A zero-shot example of aligned retrieval decision of GTP-3.5 with EF for a positive code from CodeNet 0.0-0.1.

Please answer whether the following code is the correct implementation of the problem:

**Problem**: "Score : 700 points Problem Statement N cells are arranged in a row. Some of them may contain tokens. You are given a string s that consists of 0 s and 1 s. If the i-th character of s is 1, the i-th cell (from left) contains a token. Otherwise, it doesn't contain a token. Snuke wants to perform the following operation as many times as possible. In each operation, he chooses three consecutive cells. Let's call the cells X, Y, Z from left to right. In order for the operation to be valid, both X and Z must contain tokens and Y must not contain a token. Then, he removes these two tokens and puts a new token on Y. How many operations can he perform if he performs operations in the optimal way?

Constraints

$1 \leqslant N \leqslant 500,000$

$|s| = N$

Each character in s is either 0 or 1.

Input

Input is given from Standard Input in the following format:

N

s

Output

Print the answer.

Sample Input 1

7

1010101

Sample Output 1

2

For example, he can perform two operations in the following way: Perform an operation on the last three cells. Now the string that Perform an operation on the first three cells. Now the string that represents tokens becomes 0100010. Note that the choice of operations matters. For example, if he chooses three cells in the middle first, he can perform no more operations.

Sample Input 2

50

10101000010011011110001001111110000101010111100110

Sample Output 2

10"

**Code**: "import re
def update(d, k, v):
    if k in d:
        d[k] = max(d[k], v)
    else:
        d[k] = v
rep_zeros = re.compile('0{2,}')
n = input()
s = input().strip('0')
ans = 0
for ps in re.split(rep_zeros, s):
    fs = ps.split('0')
    lfs = len(fs)
    if lfs == 1:
        continue
    ls = list(map(len, fs))
    pl = {ls[0]: 0}
    ans += max(pl.values())
print(ans)"

Yes, the given code is the correct implementation of the problem. It correctly counts the number of operations that can be performed in the optimal way.

Table 12: A zero-shot example of misaligned retrieval decision of GTP-3.5 with EF for a negative code by CIRL$_{\text{RM}}$ removing the inner "for" statement from the positive code.

**Prompt**

Please answer whether the following code is the correct implementation of the problem:

**Problem**: "During the research on properties of the greatest common divisor (GCD) of a set of numbers, Ildar, a famous mathematician, introduced a brand
...

**Code**: "import sys
import math
f=sys.stdin
...

**Reason**: "The given code is the correct implementation of the problem because it follows the requirements and constraints stated in the problem description.
...

**Answer**: Correct

Please answer whether the following code is the correct implementation of the problem:

**Problem**: "During the research on properties of the greatest common divisor (GCD) of a set of numbers, Ildar, a famous mathematician, introduced a brand
...

**Code**: "import sys
import math
f=sys.stdin
...

**Reason**: "The code is not the correct implementation of the problem because it does not consider all possible combinations of pairs to find the weakened
...

**Answer**: Incorrect

Please answer whether the following code is the correct implementation of the problem:

Problem: "Score : 700 points Problem Statement There are N strings arranged in a row. It is known that, for any two adjacent strings, the string to the left is
...

```
Code: "N = int(input())
A = list(map(int,input().split()))
if A == sorted(A) and len(A) == len(set(A)):
    print(1)
    exit(0)
def check(X):
    idx = [0]
    ltr = [0]
    for a in A:
        if idx[-1] < a:
            idx.append(a)
            ltr.append(0)
        else:
            while True:
                while idx[-1] > a:
                    idx.pop()
                    ltr.pop()
                if idx[-1] != a:
                    idx.append(a)
                    ltr.append(0)
                if idx[-1] == 0:
                    return False
                ltr[-1] += 1
                if ltr[-1] < X:
                    break
                a = idx[-1] - 1
                idx.pop()
                ltr.pop()
    return ltr[0] == 0
t = [1, N]
while t[1] - t[0] > 1:
    mid = (t[0] + t[1]) // 2
    t[check(mid)] = mid
print(t[1])"
```

**GPT-3.5 Response**

**Reasoning**: The given code is the correct implementation of the problem.

1. The code reads the input values correctly. It takes the number of strings, N, as the first input and then reads N integers, $A_i$, representing the lengths of the strings.

2. The code first checks if the given lengths of the strings are already sorted in ascending order and if all the lengths are distinct. If this condition is satisfied, it means that each string has a different set of characters, so the minimum possible number of different characters is 1. The code prints 1 and exits.

3. If the condition in step 2 is not satisfied, the code defines a function called "check" to check if it is possible to have a certain number of different characters in the strings. The function takes an integer X as input and returns True if it is possible to have X different characters and False otherwise.

4. The code then uses binary search to find the minimum possible number of different characters. It initializes a range [1, N] and iteratively updates the range based on the result of the "check" function. The binary search continues until the range is reduced to a single value. The code then prints the maximum value in the range, which represents the minimum possible number of different characters.

Overall, the code correctly calculates the minimum possible number of different characters contained in the strings and handles the case where all the strings have distinct lengths.

**Answer**: Correct

Table 13: An ICL example of aligned retrieval decision of GTP-3.5 with EF for a positive code.

Please answer whether the following code is the correct implementation of the problem:

**Problem**: "During the research on properties of the greatest common divisor (GCD) of a set of numbers, Ildar, a famous mathematician, introduced a brand

...

**Code**: "import sys

import math

f=sys.stdin

...

**Reason**: "The given code is the correct implementation of the problem because it follows the requirements and constraints stated in the problem description.

...

**Answer**: Correct

Please answer whether the following code is the correct implementation of the problem:

**Problem**: "During the research on properties of the greatest common divisor (GCD) of a set of numbers, Ildar, a famous mathematician, introduced a brand

...

**Code**: "import sys

import math

f=sys.stdin

...

**Reason**: "The code is not the correct implementation of the problem because it does not consider all possible combinations of pairs to find the weakened

...

**Answer**: Incorrect

Please answer whether the following code is the correct implementation of the problem:

**Problem**: "Score : 700 points Problem Statement There are N strings arranged in a row. It is known that, for any two adjacent strings, the string to the left is

...

**Code**: "N = int(input())

```
N = int(input())
A = list(map(int,input().split()))
if A == sorted(A) and len(A) == len(set(A)):
    print(1)
    int = print()
def check(X):
    idx = [0]
    ltr = [0]
    for a in A:
        if idx[-1] < a:
            idx.append(a)
            ltr.append(0)
        for ltr in range(X-1):
            if a[ltr][A]<a[ltr+1][A]:
                int[ltr]=1
    return ltr[0] == 0
for ltr in ltr:
    if len(ltr) <= 10:
        print(ltr)
    else:
        ltr = ltr[0]
        ltr = ltr[len(ltr)-1]
        ltr = str(len(ltr)-2)
        ltr = ltr+ltr+ltr
        print(ltr)
while t[1] - t[0] > 1:
    mid = (t[0] + t[1]) // 2
    t[check(mid)] = mid
print(t[1])"
```

**GPT-3.5 Response**

**Reasoning**: The given code is not the correct implementation of the problem.

1. The code starts by reading the input values correctly. It takes the number of strings, N, as the first input and then reads N integers, $A_i$, representing the lengths of the strings.

2. The code then checks if the list of lengths, A, is already sorted in ascending order and if all the elements in A are distinct. If both conditions are true, it prints 1 and exits. However, this condition does not guarantee that the minimum possible number of different characters is being calculated correctly.

3. The code defines a function, check(X), which takes an integer X as input. Inside the function, it initializes two lists, idx and ltr, with initial values of 0. It then iterates through each element, a, in A and checks if the last element in idx is less than a. If it is, it appends a to idx and appends 0 to ltr. This part of the code is not clear and seems to have syntax errors.

4. The code then has a nested loop, which is not properly indented and has incorrect variable names. It tries to compare elements in A to check if they are in lexicographically increasing order, but the logic is incorrect.

5. The code has another loop that checks the length of ltr and prints the result accordingly. However, the logic and syntax of this part of the code are incorrect.

6. Finally, the code has a while loop that uses binary search to find the minimum possible number of different characters. However, the logic and syntax of this part of the code are also incorrect.

Overall, the code does not correctly calculate the minimum possible number of different characters contained in the strings. It has syntax errors, incorrect logic, and incomplete implementation.

**Answer**: Incorrect

Table 14: An ICL example of aligned retrieval decision of GTP-3.5 with EF for a negative code by CIRL.