# OpenReview forum: "Intervention-Based Alignment of Code Search with Execution Feedback"
_EMNLP/2023/Conference — EMNLP 2023 Findings_

### Official Review · Reviewer_4Eqj · 2023-07-27

**Typos Grammar Style And Presentation Improvements:** NA
**Soundness:** 3

**Excitement:**

3: Ambivalent: It has merits (e.g., it reports state-of-the-art results, the idea is nice), but there are key weaknesses (e.g., it describes incremental work), and it can significantly benefit from another round of revision. However, I won't object to accepting it if my co-reviewers champion it.

**Missing References:**

NA

**Paper Topic And Main Contributions:**

This paper discusses the correctness of retrieved code snippets in code search. As discussed by the author, there are often seemingly correct code snippets retrieved since existing neural models often make decisions by token co-occurrence. To address this issue, the author proposes an RL framework to gradually change the code snippets to be wrong and consider it as a hard negative example. Besides, they propose a code intervention method by replacing sub-tress in ASTs. Experimental results on several datasets demonstrate the effectiveness of code interventions in code search.

**Questions For The Authors:**

The following questions are minor questions that I think could make me clearer about the paper:
1. Section 3.2 is quite confusing. Why could avoid computing $EF(q,c^\prime)$? Why your proposed sub-tree replacing augmentation methods are more clever and non-trivial?
2. In your experiments, you use intervened code snippets from ADV, PLM and HUMAN. Do you mean you directly take them as intervened code snippets instead of generating intervened code snippets by sub-tree replacement?
3. In Table 4, what intervened code snippets do you use to test? HUMAN and sub-tree replacement intervened codes?
4. In Table 3, I observe an MRR decrease on CSN-python. What does it mean? Does it mean that there are some wrong code snippets in CSN labeled as the true answer, and since your method could detect them hence not retrieving them, the MRR drops?

Other comments:
I think this paper has the potential to be a good work but still lacks some clarification, analysis and experiments. Besides, although I think evaluating your method on Python is enough, I strongly recommend you evaluate it on at least two other programming languages, which will make your work better.
If I find I misunderstand something, I will change my score after making it clear.

**Reasons To Accept:**

1. Considering the correctness of retrieved code snippets in code search is a very interesting and novel point to me. The internal nature of this paper is actually studying data augmentation methods. However, different from previous data augmentation approaches that focus on generating different code snippets with similar looks or semantic-preserving code snippets with dissimilar looks, this paper addresses from another perspective, the correctness of the code snippets.
2. The sub-tree replacement method for generating intervened code snippets seems quite novel.
3. Authors conduct experiments based on recent popular LLMs, which could shed light on future works.

**Reasons To Reject:**

1. The motivation for building the RL framework is not clear. As derived by the authors in Equation (4), the final policy update could be formulated as: $(EF(q,c_t)-EF(q,c_t^\prime))\partial \log p_{theta}(a_t|\hat{s_t})$. Instead of implementing an RL framework, the authors could directly optimize this loss function by just generating the intervened code snippets $c_t^\prime$. Then, we could have both $EF(q,c_t)$ and $EF(q,c_t^\prime)$. And we let the model $\theta$ to decided which one is the correct answer among set $C_t$ combined with $c_t^\prime$. The loop of re-generating intervened code snippets could also be considered as curriculum learning.
2. As I mentioned in the paper summarization, the internal nature of this paper is proposing data augmentation methods. Then, there is a need to compare the proposed augmentation methods with previous works [1,2,3].
3. Although the authors state in the 'Limitations' that they only apply the proposed method on GraphCodeBERT, there is a need to evaluate it on other code search models since the proposed method is obviously a general augmentation approach.

[1] Ding, Yangruibo, et al. "Towards learning (dis)-similarity of source code from program contrasts." arXiv preprint arXiv:2110.03868 (2021).

[2] Li, Haochen, et al. "Exploring Representation-Level Augmentation for Code Search." arXiv preprint arXiv:2210.12285 (2022).

[3] Shi, Ensheng, et al. "CoCoSoDa: Effective Contrastive Learning for Code Search." 2023 IEEE/ACM 45th International Conference on Software Engineering (ICSE). IEEE, 2023.

**Reproducibility:**

5: Could easily reproduce the results.

**Reviewer Confidence:**

4: Quite sure. I tried to check the important points carefully. It's unlikely, though conceivable, that I missed something that should affect my ratings.

---

> ### Author Rebuttal · Authors · 2023-08-29
>
> Thank you for your constructive feedback.
>
> **The loop of re-generating intervened code snippets could also be considered as curriculum learning.**
>
> We agree that at the surface-level the behavior of ours can be similar to curriculum learning, though ours, being based on an RL-based framework, outperforms such an approach (see Table A).
>
> |Method|Codeforces-valid|Codeforces-Test|CodeNet0-10| CodeNet10-20|CodeNet20-30|CodeNet30-40|
> |------|---|---|---|---|---|---|
> |Contrastive Learning w/ Simple Curriculum|86.96 |85.75 |43.86 |51.89 |55.53 |64.63|
> |CIRL|92.39|79.90|54.99 |55.98| 63.23 |66.09|
> |CIRL+CI-Human|96.74|79.02| 56.99 |57.86 |68.36 |71.04|
>
> |Method|CodeNet40-50|CodeNet50-60|CodeNet60-70|CodeNet70-80|CodeNet80-90|CodeNet90-100|
> |------|---|---|---|---|---|---|
> |Contrastive Learning w/ Simple Curriculum|72.92|77.97| 83.61 |84.94 |89.97| 93.87|
> |CIRL|78.01|80.64 |87.50 |90.81| 87.99 |98.11|
> |CIRL+CI-Human|79.32|82.44|87.75|90.94|89.79|99.06|
> **Table A. Comparison of CIRL and non-RL-based contrastive learning on Codeforces and CodeNet.**
>
>
>
> **As I mentioned in the paper summarization, the internal nature of this paper is proposing data augmentation methods. Then, there is a need to compare the proposed augmentation methods with previous works [1,2,3].**
>
> We will report this comparison in the revised draft.
>
> **Although the authors state in the 'Limitations' that they only apply the proposed method on GraphCodeBERT, there is a need to evaluate it on other code search models since the proposed method is obviously a general augmentation approach.**
>
> We agree with your point, and report the results of ContraCode+CIRL on CSN python and AdvTest.
> We will add the results of ContraCode+CIRL on Codeforces and CodeNet in the revised draft.
>
> |Model|Max. Train Epochs|AdvTest|
> |------|---|---|
> |ContraCode|10|**22.2**|
> |ContraCode+CIRL|10|22.1|
> **Table B. ContraCode+CIRL on AdvTest.**
>
> |Model|Max. Train Epochs|CSN-python|
> |------|---|---|
> |ContraCode|10|59.9|
> |ContraCode+CIRL|10|**62.2**|
> **Table C. ContraCode+CIRL on CSN-python.**
>
>
>
> **In your experiments, you use intervened code snippets from ADV, PLM and HUMAN. Do you mean you directly take them as intervened code snippets instead of generating intervened code snippets by sub-tree replacement?**
>
> Your understanding is correct.
>
> **In Table 4, what intervened code snippets do you use to test? HUMAN and sub-tree replacement intervened codes?**
>
> For the test time augmentation, we use sub-tree replacement intervened code snippets.
>
> **In Table 3, I observe an MRR decrease on CSN-python. What does it mean? Does it mean that there are some wrong code snippets in CSN labeled as the true answer, and since your method could detect them hence not retrieving them, the MRR drops?**
>
> Our conjecture is that the high lexical/structural dissimilarity between positive-negative code snippets incurs observation bias (lines 150-163) in CSN-python test set.
> This bias can be reduced in another language with smaller biases, e.g., MRR increases on CSN-ruby (see our results Table D.)
> We will add the discussion for this interesting phenomenon in the revised draft.
>
> **Other comments: I think this paper has the potential to be a good work but still lacks some clarification, analysis and experiments. Besides, although I think evaluating your method on Python is enough, I strongly recommend you evaluate it on at least two other programming languages, which will make your work better. If I find I misunderstand something, I will change my score after making it clear.**
>
> Again, thanks for constructive feedback.
> Though we could not finish all languages in time for  the rebuttal, we will continue reporting new results, during discussion and revision.
> |Model|Max. Train Epochs|CSN-ruby|
> |------|---|---|
> |GraphCodeBERT|10|70.3|
> |GraphCodeBERT+CIRL|10|**72.0**|
> **Table D. CIRL on CSN-ruby**
>
> We could address most of the requested evaluations and clarifications during the rebuttal and hope this can be reflected in your scoring.

---

### Official Review · Reviewer_Xw4P · 2023-07-28

**Soundness:** 3

**Excitement:**

3: Ambivalent: It has merits (e.g., it reports state-of-the-art results, the idea is nice), but there are key weaknesses (e.g., it describes incremental work), and it can significantly benefit from another round of revision. However, I won't object to accepting it if my co-reviewers champion it.

**Missing References:**

- "Execution-based Code Generation using Deep Reinforcement Learning" (https://arxiv.org/abs/2301.13816)
- "CodeRL: Mastering Code Generation through Pretrained Models and Deep Reinforcement Learning" (https://arxiv.org/abs/2207.01780)

**Paper Topic And Main Contributions:**

This paper aims to retrieve functionally correct code for a given natural language query in code search. Oftentimes, code search models are trained with positive code snippets having high lexical overlap with the query and then random negative code snippets that are both lexically and structurally very different from the positive code snippets. For this reason, these models often learn surface features related to syntax and lexical overlap rather than features related to semantics and functionality. To address this issue, the authors use an RL-based approach to force a model to learn from hard negative examples in which subtrees of the ASTs corresponding to positive code snippets are iteratively perturbed in ways that alter the semantics and structure. They conduct experiments with three different datasets and compare against various baselines using various techniques for obtaining negative examples.

**Questions For The Authors:**

A) In light of my comments above, could you provide some clarification on the motivation around functional correctness for code search?

B) In light of my comments above, could you provide some clarification on why RL is needed and why a simple contrastive learning approach using the new subtree perturbation method is not sufficient?

C) How does ContraCode compare to CIRL in Tables 1-4 and Figure 4? (It is only included in the first part of Table 1).

D) In Tables 2-3, why are constraints placed relating to epochs, rather than using some early stopping mechanism that yields the optimal performance for each model on the validation set?

**Reasons To Accept:**

- The approach perturbing code snippets by manipulating AST subtrees is quite interesting and could have implications for a wide variety of code-related tasks.
- The idea of using execution feedback (though not actually used in this paper) for code search is quite neat and novel.
- Though not completely novel, using execution feedback as reward in RL frameworks is not as well-explored, so it was nice to see a task formulation around that, though this feedback is not actually used in this paper.
- There are interesting code intervention baselines, particularly the CI-Human one, which might be an interesting set of results for the research community.

**Reasons To Reject:**

- I find the motivation of this work to be flawed in some ways. Namely, In Lines 119-121, the authors claim that the goal of code search is "to find functionally correct codes for each query." In fact, retrieving functionally correct code is what motivates their approach based on "execution feedback." However, I do not agree with this. Code search is generally defined as retrieving some code snippet that is relevant to the query. This is a retrieval task (i.e., retrieving a code snippet that already exists that is most relevant to the given query) rather than a generative one (i.e., generate a code snippet that implements the query). In a real-world setting, many times, it is unlikely that a code snippet that implements the exact same functionality that I am interested in exists. Nonetheless, a code snippet that does something similar or relevant (e.g., uses similar API calls) may be useful to me.  In Lines 419-422, the authors claim that the performance drop for CodeSearchNet is because the dataset is flawed; however, I would argue that the dataset is defined for a different goal than what the authors are considering.
- The use of "execution feedback" (including in the title and Figure 2) is misleading. No execution feedback signal is actually used in this work. Execution feedback is just approximated based on structural perturbations in the AST (i.e., static analysis). Based on Lines 233-238, it seems that the authors assume that any such perturbation results in a failure as execution feedback (EF = 0). In Equation 1, EF is defined in terms of whether or not the tests in a test suite pass. However, there is no guarantee that such perturbations will make the tests fail.
- The need for having a RL-based intervention algorithm is not clear and not empirically validated. Using the same subtree perturbation method to generate hard negative examples, the authors could have considered a simple contrastive learning approach, similar to ContraCode. In fact, ContraCode performs the best on the test set for Codeforces in Table 1 (contrary to what is claimed in Lines 399-401), and ContraCode is excluded in all other tables. It is not clear whether CIRL outperforms ContraCode.

**Reproducibility:**

4: Could mostly reproduce the results, but there may be some variation because of sample variance or minor variations in their interpretation of the protocol or method.

**Reviewer Confidence:**

4: Quite sure. I tried to check the important points carefully. It's unlikely, though conceivable, that I missed something that should affect my ratings.

**Typos Grammar Style And Presentation Improvements:**

- In Lines 425-426, it says that CIRL is applied to the test sets. However, CIRL involves training the underlying model. So does this mean the model is trained using the test sets?
- Lines 149-152 suggest that negative examples are always naively sampled. However, in light of ContraCode, it would be important to highlight that there is extensive work in finding hard negative examples for contrastive learning for a wide variety of retrieval tasks.

---

> ### Author Rebuttal · Authors · 2023-08-29
>
> Thank you for your constructive feedback.
>
> **I find the motivation of this work to be flawed in some ways. Namely, In Lines 119-121, the authors claim that the goal of code search is "to find functionally correct codes for each query." In fact, retrieving functionally correct code is what motivates their approach based on "execution feedback." However, I do not agree with this. Code search is generally defined as retrieving some code snippet that is relevant to the query. This is a retrieval task (i.e., retrieving a code snippet that already exists that is most relevant to the given query) rather than a generative one (i.e., generate a code snippet that implements the query). In a real-world setting, many times, it is unlikely that a code snippet that implements the exact same functionality that I am interested in exists. Nonetheless, a code snippet that does something similar or relevant (e.g., uses similar API calls) may be useful to me. In Lines 419-422, the authors claim that the performance drop for CodeSearchNet is because the dataset is flawed; however, I would argue that the dataset is defined for a different goal than what the authors are considering.**
>
> We agree that the original goal of CodeSearchNet is different from our goal of functional correctness, and in some scenarios, the original goal is sufficient. It was thus not our intention to claim the dataset is flawed, but rather argue there are other scenarios, e.g., generated codes are directly executed, where an alternative goal of functional correctness is worth pursuing.
>
> **The use of "execution feedback" (including in the title and Figure 2) is misleading. No execution feedback signal is actually used in this work. Execution feedback is just approximated based on structural perturbations in the AST (i.e., static analysis). Based on Lines 233-238, it seems that the authors assume that any such perturbation results in a failure as execution feedback (EF = 0). In Equation 1, EF is defined in terms of whether or not the tests in a test suite pass. However, there is no guarantee that such perturbations will make the tests fail.**
>
> Though reviewer Xw4P is correct in that EF may not be 0 after perturbation, there is misunderstanding that we are assuming EF=0. We are following the convention of “pseudo-labeling”, allowing and tolerating some label noises, while our subtree replacement (Section 3.3) makes noisy labeling highly unlikely. We will clarify this point in the revised draft.
>
> **The need for having a RL-based intervention algorithm is not clear and not empirically validated. Using the same subtree perturbation method to generate hard negative examples, the authors could have considered a simple contrastive learning approach, similar to ContraCode. In fact, ContraCode performs the best on the test set for Codeforces in Table 1 (contrary to what is claimed in Lines 399-401), and ContraCode is excluded in all other tables. It is not clear whether CIRL outperforms ContraCode.**
>
> We agree that a simple contrastive learning approach would behave similarly to RL-based intervention. This is not surprising, because our framework is similar to a contrastive objective if every c’ satisfies R_\theta (q,c) = R_\theta (q,c’). However, we respectfully disagree that this means RL-based framework is unnecessary—In our answer to reviewer 4Eqj, our RL-based scheduling can be replaced by a simpler curriculum learning, but RL-based approach, a principled RL approach outperforms simple heuristics. That is, the RL framework subsumes a heuristic in some setting, and also generalizes better in diverse tasks and settings.
>
> **How does ContraCode compare to CIRL in Tables 1-4 and Figure 4? (It is only included in the first part of Table 1).**
>
> CIRL outperforms ContraCode, validated in the following added evaluations on CSN, and Test time code augmentation by CIRL to CodeNet, while showing a comparable MRR score on AdvTest.
>
>
> **In Tables 2-3, why are constraints placed relating to epochs, rather than using some early stopping mechanism that yields the optimal performance for each model on the validation set?**
>
> We are already using an early stopping mechanism yielding the optimal performance. Reported results show an increase of maximum training epoch from 2 to 10.
>
> **In Lines 425-426, it says that CIRL is applied to the test sets. However, CIRL involves training the underlying model. So does this mean the model is trained using the test sets?**
>
> Our model is NOT trained using a test set, but the perturbation proposed from ours is applied to the test set.
>
> **Lines 149-152 suggest that negative examples are always naively sampled. However, in light of ContraCode, it would be important to highlight that there is extensive work in finding hard negative examples for contrastive learning for a wide variety of retrieval tasks.**
>
> We concur with your perspective concerning the significance of identifying challenging negative examples. In actuality, CIRL plays a similar role as
> hard negative mining, all while circumventing the need for extensive computational resources or memory usage.
> We will clarify that there are extensive approaches for finding hard negative mining in contrastive learning in the revised draft.

---

### Official Review · Reviewer_fqev · 2023-08-04

**Soundness:** 3

**Excitement:**

3: Ambivalent: It has merits (e.g., it reports state-of-the-art results, the idea is nice), but there are key weaknesses (e.g., it describes incremental work), and it can significantly benefit from another round of revision. However, I won't object to accepting it if my co-reviewers champion it.

**Paper Topic And Main Contributions:**

This work tackles the problem of code retrieval, specifically targeting the problem of distinguishing functionally correct code from incorrect code. They propose an approach called CIRL that introduces structural perturbations into code (e.g. replacing an entire node in the AST like an else block), where previous work used lexical changes (e.g. moving a semicolon). They apply this with a policy gradient-based RL formulation, noting that their approach allows for avoiding costly execution feedback while providing a larger edit than a simple lexical perturbation.

The paper evaluates on three categories of Python code datasets, building on GraphCodeBERT as a baseline. They demonstrate improvements in MRR over multiple code intervention baselines. Further analysis is conducted by examining the impact on retrievers after applying CIRL-based perturbations on existing test data. A small scale study is also conducted with GPT-3.5 to investigate the ability of LLMs to identify incorrect code.

**Questions For The Authors:**

* Why does ContraCode have lower validation loss but much higher test accuracy on Codeforces?
* What is the relationship between code difficulty and effectiveness of augmentation?
* The implementation of CIRL may be an obstacle to trying out the method and reproducing it – will it be made publicly available?

**Reasons To Accept:**

* The approach provides an intuitive/well-motivated improvement over previous approaches to the problem
* The approach is evaluated on multiple different benchmarks, with further breakdown by code difficulty
* Extensive implementation details are provided (in the Appendix)

**Reasons To Reject:**

* The approach is evaluated only on Python data with one code search method (GraphCodeBERT) – an advantage of lexical perturbation is the simplicity of implementation, whereas this approach seems to require a potentially complex implementation for each language. There should be some discussion around such tradeoffs and
* The LLM-baed EF investigation is not thorough and this caveat should be clear — the prompt provide a correct and incorrect example, but does not format these the same as the instruction — for ICL, the in-context examples should be structured the same as the question. The zero-shot prompt is also grammatically incorrect: “Please answer that the following code is the correct implementation of the problem” should be “Please answer whether” — this study thus needs a caveat that it is small scale and incomplete.

**Reproducibility:**

5: Could easily reproduce the results.

**Reviewer Confidence:**

2: Willing to defend my evaluation, but it is fairly likely that I missed some details, didn't understand some central points, or can't be sure about the novelty of the work.

**Typos Grammar Style And Presentation Improvements:**

* Codes -> code (code is a mass noun). Passages of code, code snippets, functions, programs…
* Unnecessary comma examples
    * 047 - remove comma
    * 093 - remove comma
    * 504 - remove comma
    * 516 - remove comma
* 046 - resulting -> resulting in, decision -> decisions
* 230 - Lexical counterfactual perturbation is an existing baseline targeted for token sequences
* 240 - "a strategy" and remove comma
* Table 1 caption — fully controlled their generation -> fully controlled in their generation process
* 515 - model decision -> model decisions
* 520 - despite -> despite that

---

> ### Author Rebuttal · Authors · 2023-08-29
>
> Thank you for your constructive feedback.
>
> **The approach is evaluated only on Python data with one code search method (GraphCodeBERT) – an advantage of lexical perturbation is the simplicity of implementation, whereas this approach seems to require a potentially complex implementation for each language. There should be some discussion around such tradeoffs**
>
> Perturbing Abstract Syntax Tree (AST) may look more complex than lexical perturbation, but straightforwardly generalizes to other languages, as AST parser (e.g. tree-sitter) easily applies to other languages for extracting a language-generic representation to perturb. We will revise to make this point more convincing and show such a generalization. See our response for reviewer 4Eqj with an additional experiment of CIRL on ruby language.
>
> **The LLM-based EF investigation is not thorough and this caveat should be clear — the prompt provides a correct and incorrect example, but does not format these the same as the instruction — for ICL, the in-context examples should be structured the same as the question. The zero-shot prompt is also grammatically incorrect: “Please answer that the following code is the correct implementation of the problem” should be “Please answer whether” — this study thus needs a caveat that it is small scale and incomplete.**
>
> We corrected grammatical errors in prompt as suggested, but our results were not sensitive to
> such change: GPT-3.5 shows a higher accuracy in positive code and CIRL, and lower accuracy than random guessing (50%) in CI-Human and CIRL_RM.
>
>
> |shot | Positive Code | Negative Code (CI-Human) | Negative Code (CIRL_RM) | Negative Code (CIRL)|
> |------|---|---|---|---|
> |zero-shot | 82.86% | 34.29% | 48.57% | 100.00%|
> |ICL | 100.00% | 17.14% | 28.57% | 100.00%|
> **Updated Figure 4.**
>
> Specifically, below is our new prompt:
>
> Please answer whether the following code is the correct implementation of the problem:
>
> Problem: <problem description>
>
> Code: <code snippet>
>
> Reasoning: <reasoning for the correctness>
>
> Answer: <Correct or Incorrect>
>
>
> For a correct code, few-shot examples for reasoning (<reasoning for the correctness>) were generated zero-shot using the following prompt:
>
> Explain why the following code is the correct implementation of the problem:
>
> Problem: <problem description>
>
> Code: <code snippet>
>
> Reason:
>
> For an incorrect code:
>
> Explain why the following code is not the correct implementation of the problem:
>
> Problem: <problem description>
>
> Code: <code snippet>
>
> Reason:
>
> Regarding the scale, we will extend to different datasets (e.g., other subsets of CodeNet, Codeforces) and models such as GPT-4.
>
>
> **What is the relationship between code difficulty and effectiveness of augmentation?**
>
> Though CIRL, with augmentation, significantly improves MRR scores regardless of code difficulty, CI-Human is effective particularly when code difficulty is high (about 2%+ of additional MRR gain in CodeNet 0-10 ~ 30-40, whereas  <2% of gain in CodeNet 40-50 ~ 90-100).
> This generality over code difficulty indicates the effectiveness of augmentation.
>
> **The implementation of CIRL may be an obstacle to trying out the method and reproducing it – will it be made publicly available?**
>
> To reduce the difficulty in reproduction, we will make all implementations publicly available.

---

### Meta-Review · Area_Chair_u3gL · 2023-09-19

**Recommendation:** 3

**Metareview:**

This paper addresses the task of code retrieval, specifically targeting the problem of distinguishing functionally correct code from incorrect code. While previous work used lexical changes in this context, this paper proposes an approach called CIRL that introduces iterative perturbations into subtrees within ASTs that alter the semantics and structure. The authors use an RL-based approach to force a model to learn from the hard negative examples created using such iterative perturbations. Experimental results on three Python code datasets demonstrate the effectiveness of code interventions in code search and show improvements in MRR over multiple code intervention baselines.

Reviewers consider this paper **overall technically sound** and identify the following **strengths** in this work:
1. *Approach*: Reviewer fqev highlights that the approach provides an intuitive and well-motivated improvement over previous approaches to the problem, and Reviewer Xw4P supports this view by adding that this approach could have implications for a wide variety of code-related tasks.
2. *Novelty*: In addition, reviewers Xw4P and 4Eqj highlight that the proposed idea and method are novel.
3. *Evaluation*: Reviewer fqev points out that the proposed approach is evaluated on multiple benchmarks, with further breakdown by code difficulty. In addition, Reviewer Xw4P notes that there are interesting code intervention baselines proposed in this work, which may be useful for the research community. Finally, Reviewer 4Eqj points out that authors conduct experiments based on recent popular LLMs, which could also be useful in future work.
4. *Level of detail*: Reviewer fqev mentions that extensive implementation details are provided in the Appendix.

At the same time, all reviewers have identified **a number of weaknesses** of the current submission and posed some clarifying questions for the authors. Specifically:
1. *Limitations to the proposed method*: As Reviewer fqev points out, the approach is evaluated only on Python data and with one code search method. In addition, the proposed approach may require a potentially complex implementation for each language (more so than the simpler lexical perturbation approach) but these aspects are not currently discussed in the paper. Reviewer 4Eqj agrees with this view and also suggests that the presented method be evaluated with other code search models and using other programming languages.
2. *Weaknesses related to the experiments with LLMs*: Reviewer fqev identifies several weaknesses in these experiments – for more details, see their review.
3. *Justification of certain methodological decisions*: Reviewer Xw4P poses a number of clarifying questions about implementation and methodological decisions – see their review. Reviewer 4Eqj supports this view by posing similar questions about the motivation behind using the RL framework.

In addition, all reviewers have provided further suggestions for improvements and identified some missing references. The authors did a good job addressing reviewers' feedback and acknowledged reviewers' suggestions. These will, hopefully, be reflected in the revised version of the paper.

---

### Decision · Program_Chairs · 2023-10-07

**Decision:**

Accept-Findings

**Comment:**

This paper addresses the task of code retrieval, specifically targeting the problem of distinguishing functionally correct code from incorrect code. While previous work used lexical changes in this context, this paper proposes an approach called CIRL that introduces iterative perturbations into subtrees within ASTs that alter the semantics and structure. The authors use an RL-based approach to force a model to learn from the hard negative examples created using such iterative perturbations. Experimental results on three Python code datasets demonstrate the effectiveness of code interventions in code search and show improvements in MRR over multiple code intervention baselines.

Reviewers consider this paper **overall technically sound** and identify the following **strengths** in this work:
1. *Approach*: Reviewer fqev highlights that the approach provides an intuitive and well-motivated improvement over previous approaches to the problem, and Reviewer Xw4P supports this view by adding that this approach could have implications for a wide variety of code-related tasks.
2. *Novelty*: In addition, reviewers Xw4P and 4Eqj highlight that the proposed idea and method are novel.
3. *Evaluation*: Reviewer fqev points out that the proposed approach is evaluated on multiple benchmarks, with further breakdown by code difficulty. In addition, Reviewer Xw4P notes that there are interesting code intervention baselines proposed in this work, which may be useful for the research community. Finally, Reviewer 4Eqj points out that authors conduct experiments based on recent popular LLMs, which could also be useful in future work.
4. *Level of detail*: Reviewer fqev mentions that extensive implementation details are provided in the Appendix.

At the same time, all reviewers have identified **a number of weaknesses** of the current submission and posed some clarifying questions for the authors. Specifically:
1. *Limitations to the proposed method*: As Reviewer fqev points out, the approach is evaluated only on Python data and with one code search method. In addition, the proposed approach may require a potentially complex implementation for each language (more so than the simpler lexical perturbation approach) but these aspects are not currently discussed in the paper. Reviewer 4Eqj agrees with this view and also suggests that the presented method be evaluated with other code search models and using other programming languages.
2. *Weaknesses related to the experiments with LLMs*: Reviewer fqev identifies several weaknesses in these experiments – for more details, see their review.
3. *Justification of certain methodological decisions*: Reviewer Xw4P poses a number of clarifying questions about implementation and methodological decisions – see their review. Reviewer 4Eqj supports this view by posing similar questions about the motivation behind using the RL framework.

In addition, all reviewers have provided further suggestions for improvements and identified some missing references. The authors did a good job addressing reviewers' feedback and acknowledged reviewers' suggestions. These will, hopefully, be reflected in the revised version of the paper.